# Cross-Attention-Guided Feature Alignment Network for Road Crack Detection

**Chuan Xu** [1,†]**, Qi Zhang** [1]**, Liye Mei** [1,†]**, Xiufeng Chang** [2]**, Zhaoyi Ye** [1]**, Junjian Wang** [3]**, Lang Ye** [3] **and Wei Yang** [3,*]

[1] School of Computer Science, Hubei University of Technology, Wuhan 430068, China; 20200064@hbut.edu.cn (C.X.); 102111136@hbut.edu.cn (Q.Z.); meiliye@hbut.edu.cn (L.M.); 102101051@hbut.edu.cn (Z.Y.)
[2] Unit 92493, Huludao 125000, China; 00031417@whu.edu.cn
[3] School of Information Science and Engineering, Wuchang Shouyi University, Wuhan 430064, China; 2003131011@wsyu.edu.cn (J.W.); yelang@wsyu.edu.cn (L.Y.)
[*] Correspondence: yangwei403@wsyu.edu.cn
[†] These authors contributed equally to this work.

**Abstract:** Road crack detection is one of the important issues in the field of traffic safety and urban planning. Currently, road damage varies in type and scale, and often has different sizes and depths, making the detection task more challenging. To address this problem, we propose a Cross-Attention-guided Feature Alignment Network (CAFANet) for extracting and integrating multi-scale features of road damage. Firstly, we use a dual-branch visual encoder model with the same structure but different patch sizes (one large patch and one small patch) to extract multi-level damage features. We utilize a Cross-Layer Interaction (CLI) module to establish interaction between the corresponding layers of the two branches, combining their unique feature extraction capability and contextual understanding. Secondly, we employ a Feature Alignment Block (FAB) to align the features from different levels or branches in terms of semantics and spatial aspects, which significantly improves the CAFANet's perception of the damage regions, reduces background interference, and achieves more precise detection and segmentation of damage. Finally, we adopt multi-layer convolutional segmentation heads to obtain high-resolution feature maps. To validate the effectiveness of our approach, we conduct experiments on the public CRACK500 dataset and compare it with other mainstream methods. Experimental results demonstrate that CAFANet achieves excellent performance in road crack detection tasks, which exhibits significant improvements in terms of F1 score and accuracy, with an F1 score of 73.22% and an accuracy of 96.78%.

**Keywords:** road crack detection; multi-scale features; cross-layer interaction; feature alignment

## 1. Introduction

Roads are of vital importance to economic growth and development, offering substantial societal advantages. They enable mobility and connectivity, granting individuals convenient access to employment opportunities, social services, healthcare, and educational facilities [1]. Road infrastructure is an integral component of public assets. However, traffic accidents can occur due to the gradual wear and deterioration of the road surface. This can be attributed to various factors including location, traffic volume, weather conditions, and construction materials. Many countries already account for 99% of their total road mileage through road maintenance. Hence, Road Crack Detection (RCD) holds significant importance in road infrastructure as it entails road damage identification and classification [2–5]. Identifying roads that require maintenance to mitigate potential safety hazards is of paramount importance for ensuring cost-effective road maintenance and enhancing traffic safety. In recent years, numerous experts and scholars have devoted themselves to

researching automatic RCD and have obtained promising research results. At present, automatic RCD research is roughly divided into three methods: traditional image processing methods, machine learning methods, and deep learning methods.

In traditional methods, the detection of road damage regions typically relies on the threshold-based method [6–8]. These algorithms employ different thresholds to quickly identify the results in the input image. Using this method, one can easily spot road damage, since it tends to absorb more light than other parts of the image, resulting in darker areas. Nevertheless, in the presence of noise, pixels with lower intensity than the damaged pixels can significantly compromise performance of overall detection. Having no comprehensive understanding of global information makes these methods susceptible to noise and heavily dependent on threshold selection. Other researchers have explored the use of feature descriptors that are designed artificially to recognize damage in road images. As an example, Gabor filters [9] and wavelet transforms [10] have been instrumental in making remarkable advancements in detecting simple cracks. Despite these advancements, challenges persist due to the complexity of road damage, which exhibits diverse topologies, arbitrary shapes, and varying widths and can be mixed with other strong disturbances such as oil spots, weeds, and stains on the road. As a result, the current performance of these methods remains limited in accurately detecting and categorizing all types of road damage effectively.

The machine learning method has seen significant progress and widespread adoption in the realm of RCD. In order to detect cracks more accurately and efficiently, researchers have explored a variety of approaches. For instance, an enhanced active contour model and a greedy search-based Support Vector Machine (SVM) have been applied to study bridge damage detection [11]. Ai [12] proposes another approach to calculate probability maps that integrate neighborhood information at multiple scales using an SVM. The Probabilistic Generation Model (PGM) and SVM probability maps are merged using a fusion algorithm, enabling crack detection with greater accuracy than individual probability maps. Using the random forest method, Prasanna et al. [13] classify multiple spatially adjusted visual features. In contrast, these detection methods are constrained by their ability to only identify learned cracks, making them less effective at detecting new or previously unseen cracks. To address these limitations, CrackForest [14] is introduced, which uses a randomly structured forest to detect cracks automatically. By meticulously choosing crack features and comprehensively understanding their internal structure, this method proficiently suppresses noise. Nevertheless, CrackForest does not account for different kinds of damage in complex crack extraction scenarios. In traditional methods, cracks are often simulated by manually setting color or texture features, but these features are not robust enough to handle changes in diverse environments. Consequently, in the context of complex road images, manually designed features demonstrate limited effectiveness in extracting cracks. To enhance crack detection performance in such varying conditions, there is a need for more adaptive and data-driven approaches that can automatically learn and generalize features from the input data.

Recent theoretical progress reveals that deep learning can proficiently tackle intricate problems by autonomously acquiring features at various hierarchical levels [15]. Deep Convolutional Neural Networks (DCNNs) are known for their ability to extract rich hierarchical features [16–21], and their end-to-end trainable framework of deep learning has demonstrated remarkable advancements in pixel-level semantic segmentation tasks [22–24]. As a result, several crack detection methods leveraging deep learning, such as object detection [25,26] and image block segmentation [27–29], have emerged. However, these approximate methods lack the ability to capture intricate crack details at the pixel level, resulting in imprecise categorization and assessment of crack types and severity levels in subsequent stages. To tackle this issue, Huang et al. [30] propose an approach employing the FCN to extract cracks at the pixel level. Nevertheless, this method fails to consider that cracks of varying widths and topologies necessitate context information of different sizes. In addition, it neglects to consider the different impacts of various crack features

on crack detection, treating all features on the same level. There are other studies in the literature [31,32] that use 3D crack detection networks based on DCNNs to detect pixel-level cracks in 3D asphalt pavements. However, the uniform convolution kernels used in the network's convolutional layer may confuse the target and context of the analysis. With the SegNet encoder–decoder architecture, Zou et al. [33] implement DeepCrack. They combine the convolution features from both the encoder and decoder networks at the same scale to achieve accurate pixel-level crack detection. However, the learning mechanisms during the encoding–decoding stage of the SegNet network are relatively simplistic, and the up-sampling in shallow layers does not entirely recover spatial information. Song et al. [34] propose a crack segmentation network based on the DeepLabv3 [35] framework, which successfully achieves accurate pixel-level segmentation of tunnel cracks. Despite the method's successful utilization of the Atrous Spatial Pyramid Pooling (ASPP) [36] module to capture multi-scale information, it does not fully recognize the significance of the up-sampling operation in improving detection results. Researchers are still working on robust pixel-level crack detection in trainable DCNN models that incorporate rich semantic information, although deep learning methods yield superior results compared to traditional methods. Based on DCNNs, the crack detection methods mentioned above fail to properly classify cracks and assess damage severity.

The utilization of deep learning features in these detection methods greatly enhances the detection performance of RCD. However, they still face some significant issues to solve. First, many types of road damage, such as cracks, potholes, broken shoulders, and roadside facilities, exist. Each damage type possesses unique characteristics and manifestations, making it difficult to learn effective multi-scale characteristics for achieving efficient RCD. Second, road damage occurs in various sizes and shapes, highlighting the importance of utilizing multiple feature maps with different resolutions to capture features at diverse scales. Third, deep learning models need to have good generalization performance and can be applied to images from different geographical regions, different road types, and different lighting conditions, which requires the model not only to fit specific data sets but also to have the ability to generalize to new situations.

Taking these issues into account, we propose a cross-attention-guided feature alignment network specifically designed for RCD. Firstly, the Cross-Layer Interaction (CLI) module [37] is utilized to effectively capture multi-scale spatial information, enrich the feature space, establish long-term dependencies between diverse channels, and refine feature extraction. This module enhances the information interaction and correlation among multi-scale features, thereby improving the accuracy of feature fusion and effectively addressing scale variations. Secondly, the obtained multi-scale features are fed into the Feature Alignment Block (FAB) [38], which aids the network in better understanding the semantic information in road images. This module enhances the focus on road damage areas, reduces noise interference, improves the feature representation capability of small targets, and achieves precise RCD. Lastly, the fused features are processed through the segmentation detection module to generate the final segmentation results. In conclusion, this research makes the following key contributions:

(1) By incorporating the CLI module into the backbone network, it promotes the interaction between global and local information, thereby enhancing CAFANet's ability to express multi-scale features.

(2) The introduction of the FAB ensures the consistency in scale and semantics between different levels or branches of features, improving CAFANet's detection capability for small targets.

(3) We extensively conduct experiments on the well-known CRACK500 dataset for road crack detection, further validating the superiority of CAFANet.

There are four remaining sections in this paper: Section 2 provides an overview of the network's architecture, including the CLI module and the FAB, and explains how they contribute to feature extraction and fusion. Section 3 describes the dataset and experimental

settings, as well as provides comprehensive results. Section 4 delves into the generalization of the experiments, while Section 5 summarizes this work.

## 2. Materials and Methods

### 2.1. Network Architecture

Road crack detection can be modeled as an image segmentation problem, aiming to accurately separate the damaged areas from the background in road images. Given a road image as input, the goal is to automatically segment the damaged regions from the background using algorithms or models. The segmentation results are usually represented by pixel-level segmentation masks, where pixels belonging to the damaged areas are labeled as foreground, and background pixels are labeled as background. Adopting an encoder–decoder structure is a common design for end-to-end segmentation networks. The encoder is responsible for extracting high-level semantic features from the input image, while the decoder, through step-wise up-sampling and feature fusion operations, restores the features extracted by the encoder to the original image size and generates the segmentation result. However, performing feature fusion in the decoder is a challenge, as effectively integrating features of different resolutions and preserving detailed information are crucial for accurate segmentation results. Additionally, due to the potential scale variations in road damage regions, the network needs to be capable of adapting to different scales to accurately segment damage of various sizes.

Based on the above idea, we propose a cross-attention-guided feature alignment network specifically designed for road crack detection. Figure 1 illustrates the overall network framework. To accurately detect road damage, we construct a multi-stage architectural model consisting of two layers. In the feature extraction stage, we employ four feature levels to extract rich features from the images, generating feature maps at different scales. Each feature hierarchy comprises Multi-Scale Patch Embedding (MSPE) and Multi-Head Convolutional self-Attention (MHCA) blocks to enhance feature representation. We utilize the CLI module to establish interaction between different feature levels, attentively incorporating the two types of feature information. Subsequently, we fuse the extracted features at different scales by using the FAB. Ultimately, we obtain the RCD result.

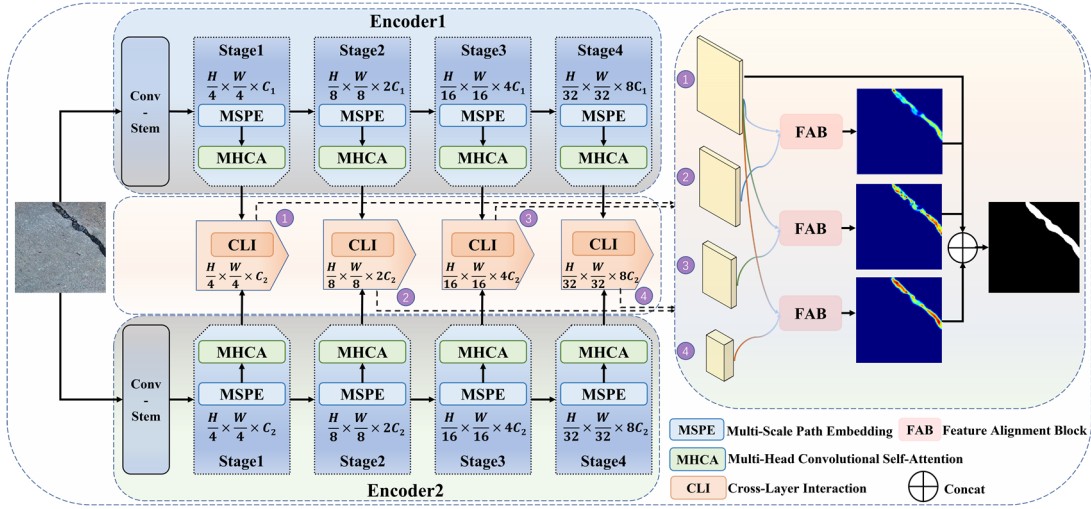

**Figure 1.** Overview of the proposed CAFANet.

### 2.2. Cross-Layer Interaction

Road cracks often occur on complex backgrounds, such as road texture, changing lighting conditions, shadows, and road markings. These background disturbances pose difficulties for crack detection because they resemble background features and make accurate distinction challenging. Moreover, road cracks exhibit different shapes (e.g., linear, reticular, speckled) and sizes (ranging from a few millimeters to several meters). This

diversity complicates the precise detection of cracks, necessitating the processing of crack instances of various shapes and sizes. MSPE and MHCA blocks possess unique feature extraction and context understanding abilities, capturing rich multi-scale feature information. Simultaneously, using the CLI module to attentively focus on the extracted feature information helps the proposed model better capture the contextual information of road cracks, thereby enhancing CAFANet's accuracy.

Feature Extraction. With an original image P of $H \times W \times 3$ in size as input, we pass it through encoder1 and encoder2 simultaneously for feature extraction. In encoder1, we utilize a trunk block consisting of two $3 \times 3$ convolutional layers with a stride of 2. In the first convolutional layer, there are channels $\frac{C_1}{2}$, and in the second layer, there are channels $C_1$. A feature of size $\frac{H}{4} \times \frac{W}{4} \times C_1$ is generated as a result of this operation, with $C_1$ representing the channel size in stage 1. After each convolutional operation, we apply batch normalization [39] and activate the output using the Hardswish function [40]. We stack the MSPE and MHCA blocks in each stage, ranging from stage 1 to stage 4. Figure 2 shows that we first employ an MSPE block, which introduces coarse-grained and fine-grained visual tokens at the same time. We employ overlapping patches in convolution operations to achieve this. Specifically, we input a token map (i.e., 2D-reshaped output feature map) from a previous stage into each subsequent stage. Through the convolutional patch embedding layer, we can modify the stride and padding parameters to adjust the length of token sequences. By employing varying patch sizes, it becomes viable to generate features of identical dimensions (i.e., resolution). Consequently, we construct multiple layers with different kernel sizes in parallel for the convolutional patches. As a result of encoder1 and encoder2's multi-path structures, which have more embedding layers, we employ depth-wise separable convolution in the MHCA block to reduce model parameters and computational costs. In this convolution, the depth-wise convolution has a kernel size of $1 \times 1$, and the point-wise convolution has a kernel size of $1 \times 1$. After each convolutional layer, we apply batch normalization. Additionally, the Hardswish activation function is used. Next, we independently feed the token embedding features of different sizes into each MHCA encoder. Finally, we utilize a depth-wise residual bottleneck block, which consists of a $1 \times 1$ convolution combined with the same channel size, a $3 \times 3$ depth-wise convolution, and a $1 \times 1$ convolution coupled with a residual connection. After performing the above operations, stage 1 to 4 in encoder1 respectively generate a map of features with a size of $64 \times 64$, $32 \times 32$, $16 \times 16$, and $8 \times 8$ from the input images, with separation rates of $\frac{1}{4}$, $\frac{1}{8}$, $\frac{1}{16}$, and $\frac{1}{32}$. Furthermore, the four feature maps have 64, 128, 256, and 512 channels, respectively. Similarly, the original image is also fed into encoder2, and the same feature extraction process as in encoder1 is carried out. The four stages in encoder2 produce feature maps with dimensions of $64 \times 64$, $32 \times 32$, $16 \times 16$, and $8 \times 8$, and the number of channels is 128, 256, 512, and 1024. We represent the four feature maps in encoder1 as $[P_1, P_2, P_3, P_4]$ and denote the four feature maps in encoder2 as $[P_1', P_2', P_3', P_4']$.

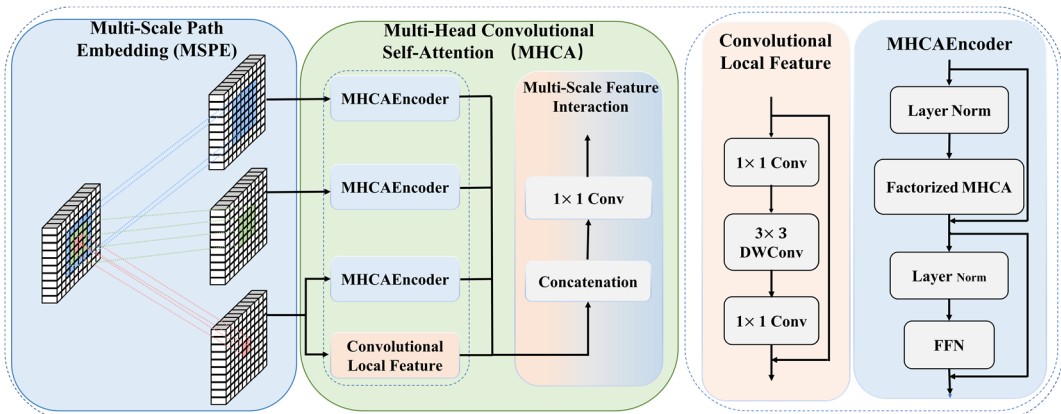

**Figure 2.** Architecture of the multi-scale path embedding and multi-head convolutional self-attention.

Feature Interaction. Figure 3 shows the feature-crossing structure of the CLI module. We cross-extract features obtained from the four stages of encoder1 and encoder 2 in a one-to-one correspondence, that is, $P_1$ and $P'_1$, $P_2$ and $P'_2$, $P_3$ and $P'_3$, $P_4$ and $P'_4$. Here are the specific operations involved in feature interaction: first, the input feature matrices with shapes $[B, C, H, W]$ and $[B, 2C, H, W]$ are reshaped separately. This transformation converts them from 4D tensors to 3D tensors, and we exchange their second and third sizes to obtain tensors $e$ and $r$. We simultaneously perform normalization and global averaging pooling on the tensors $e$ and $r$ to obtain feature vectors of length 1, which are denoted as $e\_t$ and $r\_t$. We map $e\_t$ and $r\_t$ to the dimensions of each other and insert a dimension of length 1 on the second dimension. We then concatenate $r$ and $e\_t$, $e$ and $r\_t$ in the second dimension, and subsequently feed the concatenated feature tensors into the transformer encoder for their individual cross-attention calculation.

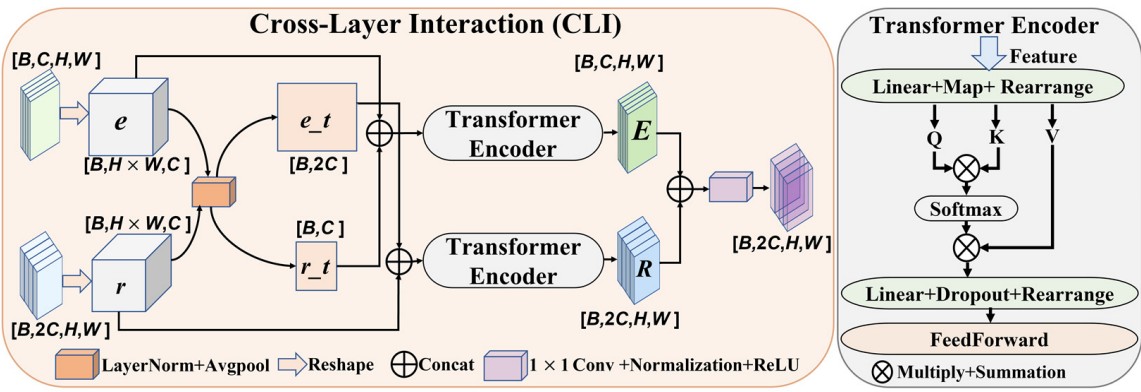

**Figure 3.** Architecture of the cross-layer interaction.

As a result of the study of human visual function, the attention mechanism [41] is initially applied to machine translation. This algorithm has been extensively used in recognizing speech, images, and natural language over the years. By directing its attention resources towards the target region, it focuses on gathering detailed information while suppressing irrelevant data. The attention mechanism in deep learning involves assigning different weights to inputs or features. The attention mechanism gives more weight to important information and assigns less weight to relatively unimportant information. In the attention mechanism workflow, the first step involves scoring the hidden state of each unit in the encoder. Subsequently, the scores are normalized using the SoftMax function [42] to derive the corresponding weights. For the next layer, to obtain the input variable, we perform a summation across all time steps, where each hidden state is multiplied by its corresponding weight and then summed. This weighted summation process ensures that the important information from the previous layer is effectively integrated into the subsequent layers. In transformer encoder, the input feature is mapped to three spaces, Query ($Q$), Key_i ($K_i$) and Value_i ($V_i$), by linear transformation, and a series of <Key, Value > data pairs are written as Sou. We can divide the process of computing attention into three stages.

In the first stage, the Einstein sum function and computation mechanism can calculate the similarity or correlation between the two, based on the $Q$ and a $K_i$. It is as follows:

$$Similarity(Q, K_i) = Q \times K_i \tag{1}$$

In the second stage, the calculation method of the SoftMax function to convert the scores of the first stage is introduced. From one perspective, we can normalize the initially calculated scores and transform them into a probability distribution, ensuring that the weights of all elements sum up to 1. From another perspective, the SoftMax mechanism

enables us to accentuate the weights of crucial elements. It is common practice among researchers to utilize the following formula for computation:

$$b_i = SoftMax(Sim_i) = \frac{a^{Sim_i}}{\sum_{j=1}^{L_x} a^{Sim_j}} \tag{2}$$

where $L_x = \| Sou \|$. The computation in the second stage yields coefficient weights $b_i$ corresponding to $V_i$, and then weighted summation can obtain the attention value:

$$Atten(Q, Feature) = \sum_{i=1}^{L_x} b_i \times V_i \tag{3}$$

In the above formula, $i$ represents the index dimension, and $a^{Sim_i}$ is a dot product matrix at the index dimension $i$. Through the calculation of the above three stages, we can obtain the attention value for $Q$.

We take out all elements except the first element and reshape them into a 4D tensor and acquire the feature map of the feature interaction results and record the two results as $E$, $R$. We concatenate $E$ and $R$ in the second dimension and through two layers of convolution, normalization, and ReLU function activation series of operations. Finally, after going through the CLI module, we receive the feature mapping of $P_1$ and $P_1'$, $P_2$ and $P_2'$, $P_3$ and $P_3'$, $P_4$ and $P_4'$ for their respective interactions. The sizes of the four feature maps are $64 \times 64$, $32 \times 32$, $16 \times 16$, $8 \times 8$, and the number of channels is 128, 256, 512, 1024.

### 2.3. Feature Alignment Block

Crack images display considerable variations owing to variations in lighting conditions, surface types, and background textures. At different scales, they manifest diverse features: larger scales enable reliable crack detection while exhibiting poorer localization, potentially overlooking thin cracks. Within small scales, it preserves the detail, but the clutter in the background texture significantly impacts detection. Incorporating the feature alignment operation into the fusion module helps the network focus more on the crack region, reduces noise interference, enhances the feature expression capability, and improves the network's understanding of semantic information in road images, thereby achieving accurate crack detection.

Before entering the FAB, we first initiate the adjustment of channel sizes for the four feature maps obtained from the CLI module by applying a convolutional layer with a kernel size of 3. Subsequently, a series of concatenation and convolution operations are performed to achieve channel sizes of 256, 512, 1024, and 2048. However, the four feature maps still have the following sizes: $64 \times 64$, $32 \times 32$, $16 \times 16$, $8 \times 8$, and we name the four feature maps $[P_5, P_6, P_7, P_8]$. Then, to achieve a channel reduction to 256 in $[P_6, P_7, P_8]$, we utilize a convolutional layer with a kernel size of 3. Simultaneously, we up-sample after dimensionality reduction, adjusting all sizes of $[P_6, P_7, P_8]$ to 64. Therefore, we also obtain 4 new feature maps and denote them as $[P_5, P_6', P_7', P_8']$.

This paper shows the fusion process of feature alignment in Figure 4. In the FAB, using a convolutional layer with a 1 m wide kernel, we obtain the input feature map $[M1, N1]$ with $C$ channels. Subsequently, we increase the number of channels to $2C$ in the feature map $[M2, N2]$. Then, $[M2, N2]$ is spliced in the second dimension to obtain the feature map $F1$ with the number of channels of $4C$. The next step is to reduce the number of channels of the feature map $F1$ from $4C$ to $2C$ by using a convolution with convolution kernel size of 1, and the feature map $F2$ is available here. In order to distinguish different spatial positions, we adopt a convolution of $1 \times 1$ to conduct spatial screening of $F2$. The SoftMax function is applied to the feature map $F2$, resulting in a channel size of 2 for the obtained feature map $F3$. At the same time, with weighted different feature channels, we also utilize a $1 \times 1$ convolution to screen $F2$ channels and acquire the feature map $F4$ with the number of channels $C$ through a sigmoid function [43]. The feature map $F5$ is obtained by multiplying the first channel of feature map $F3$ with elements in the corresponding

position of feature map *F4*, and feature map *F5* indicates the probability of the first channel. The feature map *F6* is acquired by multiplying the remaining channels in feature map *F3* with elements in the corresponding positions in feature map *F4*, and the feature figure *F6* represents the probability of the first channel. Weighted fusion is performed on feature map *F5* to combine with input feature map *M1* and on feature map *F6* to combine with input feature map *N1*, resulting in the feature map [*M3*, *N3*]. In the end, we concatenate feature maps *M3* and *N3* along the first dimension to obtain the output feature map *F7*. According to the feature alignment operation of the FAB, we input the feature maps [$P_5$, $P_6'$], [$P_5$, $P_7'$], and [$P_5$, $P_8'$] into the FAB, respectively, and finally obtain three new feature maps, denoted as [$P_6''$, $P_7''$, $P_8''$].

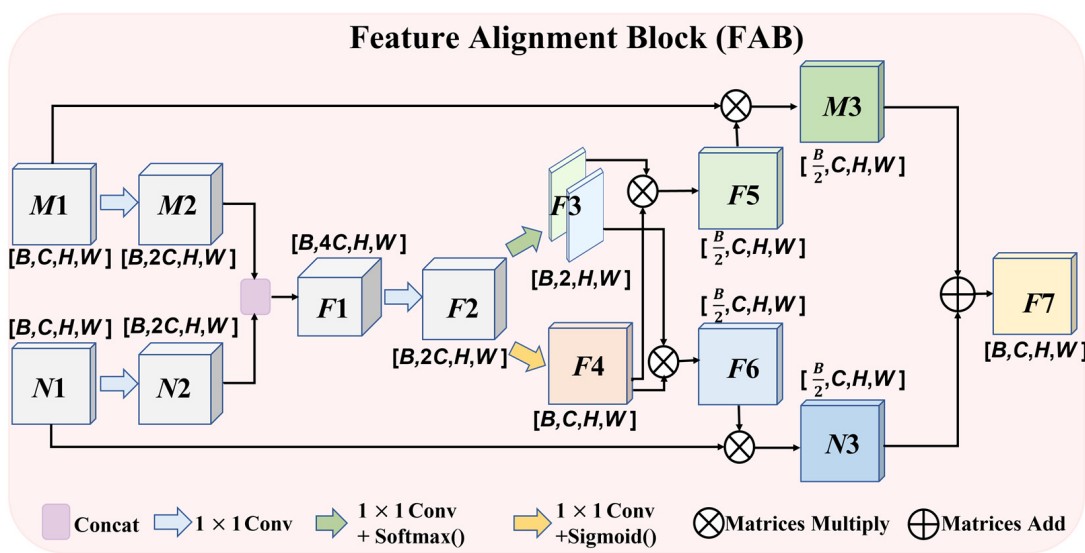

**Figure 4.** Architecture of the feature alignment block.

### 2.4. Loss Function

In road crack detection tasks, the normal road regions often occupy a larger proportion in the image, while the damage regions are relatively smaller. This leads to class imbalance, making it challenging for the model to effectively learn features of the damaged regions. In the road crack detection task, accurate localization and segmentation of the damage regions are required. Focal Loss [44] adjusts the losses' function weights so that they focus more on samples that are difficult to classify, i.e., the damaged regions. This enhances the model's learning ability for the damage regions and improves the detection performance. Dice Loss [45] considers the overlap between the predicted results and the ground truth labels during the computation, directing the model's attention towards accurate pixel classification and boundary matching. Compared to other loss functions, Dice Loss is more sensitive to learning boundary pixels and can improve the segmentation quality of the damage regions. To address class imbalance and segmentation accuracy, we introduce a flexible optimization approach by combining Focal Loss and Dice Loss. This hybrid loss function allows the proposed method to effectively tackle the challenges associated with imbalanced classes while improving the precision of the segmentation.

We employ the Focal Loss function to address the extreme class imbalance between positive and negative samples, thus enhancing the binary cross-entropy loss with the development of Focal Loss. It is a cross-entropy loss that is dynamically scaled with a dynamic scaling factor. During training, it adaptively reduces the weights of easily separable samples, allowing the weights to concentrate more rapidly on difficult-to-separate samples. The formula is as follows:

$$L_{Focal} = -\sigma(1 - T)^\varsigma \log(T) \qquad (4)$$

In this formula, $\sigma$ is the balancing factor that helps address class imbalance. The parameter $\varsigma$ controls the focusing effect, allowing the loss to pay more attention to hard-to-classify examples. The term $(1-T)^\varsigma$ serves as the dynamic scaling factor. It down-weights well-classified examples (where p is close to 1) and assigns higher weights to misclassified examples (where $T$ is closer to 0). By incorporating the dynamic scaling factor, Focal Loss encourages the model to focus more on difficult samples, effectively addressing the issue of imbalanced classes and improving the model's ability to handle challenging cases.

Using the Dice coefficient, we can calculate the similarity between two samples using the Dice Loss function to address foreground–background imbalance. This formula is as follows:

$$L_{Dice} = 1 - \frac{2|M \cap N|}{|M|+|N|} \tag{5}$$

In this formula, the $|M \cap N|$ expresses the number of pixels where both samples have a positive value. The sum of $|M|$ and the sum of $|N|$ represent positive pixels totaled in their respective samples. Keeping the Dice Loss to a minimum, we encourage the model to achieve a higher overlap between the foreground and background, indicating a better similarity between the samples. In this way, the segmentation task is more accurate as it balances foreground and background classes.

Based on the calculations of the aforementioned loss functions, we can represent the mixed loss used as follows:

$$L_{Mixed} = \lambda \times L_{Focal} + (1-\lambda) \times L_{Dice} \tag{6}$$

In this formula, $\lambda$ represents the coefficient that determines the balance between the two losses. By adjusting the value of $\lambda$, we can control the emphasis given to each component of the mixed loss. The mixed loss combines the strengths of Focal Loss and Dice Loss, addressing the issues of class imbalance and segmentation accuracy. By optimizing this mixed loss during training, CAFANet's goal is to improve road crack detection performance.

### 2.5. Evaluation Metrics

Eight evaluation metrics were selected to quantitatively assess the performance of CAFANet, including *Precision*, *Recall*, *F1 score* [46], *Intersection Over Union_0 (IOU_0)*, *Intersection Over Union_1 (IOU_1)*, *Mean Intersection Over Union (mIOU)*, *Overall Accuracy (OA)*, and *Kappa* coefficient. Specifically, we define the evaluation metrics as follows:

(1) *Precision*: This statistic indicates the percentage of samples predicting positive results that are also positive in reality. It measures the classifier's accuracy when predicting a positive sample. By using Equation (7), we can calculate this metric:

$$Precision = \frac{TP}{TP + FP} \tag{7}$$

(2) *Recall*: When an actual positive sample is produced, it indicates the percentage of positive samples correctly predicted by the classifier. It measures the classifier's ability to cover positive samples. Due to this, it is also referred to as sensitivity, which can be calculated using Equation (8):

$$Recall = \frac{TP}{TP + FN} \tag{8}$$

(3) *F1 score*: The combination of Precision and Recall rates comprises this metric. A higher F1 score indicates a more accurate classifier. This metric can be calculated using Equation (9):

$$F1 = \frac{2 \times TP}{2 \times TP + FP + FN} \tag{9}$$

(4) *IOU_0*: The first category label occurring simultaneously rather than other segmentation results with truth. It measures the segmentation accuracy of the classifier for the first category. This metric can be calculated using Equation (10):

$$IOU\_0 = \frac{TN}{TN + FN + FP} \tag{10}$$

(5) *IOU_1*: The second category of segmentation results is the ratio with the true label. It measures the accuracy of the second category segmentation. This metric can be calculated using Equation (11):

$$IOU\_1 = \frac{TP}{TN + FN + FP} \tag{11}$$

(6) *mIOU*: It compares the segmentation results of all categories and the real label. It synthesizes the segmentation accuracy of each category and the segmentation performance is evaluated using this method. This metric can be calculated using Equation (12):

$$mIOU = \frac{IOU\_0 + IOU\_1}{2} \tag{12}$$

(7) *OA*: It indicates the classification accuracy of the classifier on all samples. It measures the overall ability of a classifier to classify all categories. This metric can be calculated using Equation (13):

$$OA = \frac{TP + TN}{TP + TN + FP + FN} \tag{13}$$

(8) *Kappa*: A measure of consistency for an observer or classifier. It can eliminate the effect of accidental consistency and provide a more accurate measure for evaluating the consistency of a classifier or observer. This metric can be calculated using Equations (14) and (15):

$$TMP = \frac{(TP + FP)(TP + FN) + (FP + TN)(FN + TN)}{(TP + TN + FP + FN)^2} \tag{14}$$

$$Kappa = \frac{OA - TMP}{1 - TMP} \tag{15}$$

In the provided Equations (7)–(15), TP samples are those which are correctly predicted to be positive. TN is the number of true negative samples, which are correctly predicted to be negative. FP indicates the number of false positive samples predicted as positive but subsequently found not to be. FN refers to the number of false negative samples, which are incorrectly predicted to be negative while in fact being positive.

### 2.6. Parameter Settings

To build all of the models, we use the PyTorch framework. We utilize an NVIDIA Tesla A100 GPU with 80 gigabytes of onboard memory in the experiment. In the network model, we set 4 as the batch training size when configuring the training parameters. The initial learning rate is 0.0035. Using the training sets, we train the framework to generate the "optimized" model. We halt the training process if the training losses show no reduction for 500 consecutive epochs in order to mitigate overfitting. In the process, when the training dice scores reach their highest point, we save the generator model weights. In the test set, we use the model that performs best in the validation set.

## 3. Experiments and Results

### 3.1. Datasets

To demonstrate the superiority of CAFANet, we employ the public CRACK500 dataset [47] due to the difficulty in obtaining RCD data. The dataset consists of approximately 500 images, each with a resolution of approximately 2000 × 1500. These images

are obtained from the main campus of Temple University with a phone. We divide each image into 16 non-overlapping regions and save only regions with over 1000 crack pixels due to computing resource limitations. Each crack image undergoes pixel-level annotation. Therefore, the CRACK500 test data consist of 1124 crack images, 1896 crack images for the training data, and 348 crack images for the validation data. Figure 5 shows some data samples, with the first row showing the original images and the second row showing the corresponding label images. We mark the crack region in the original image with a red box.

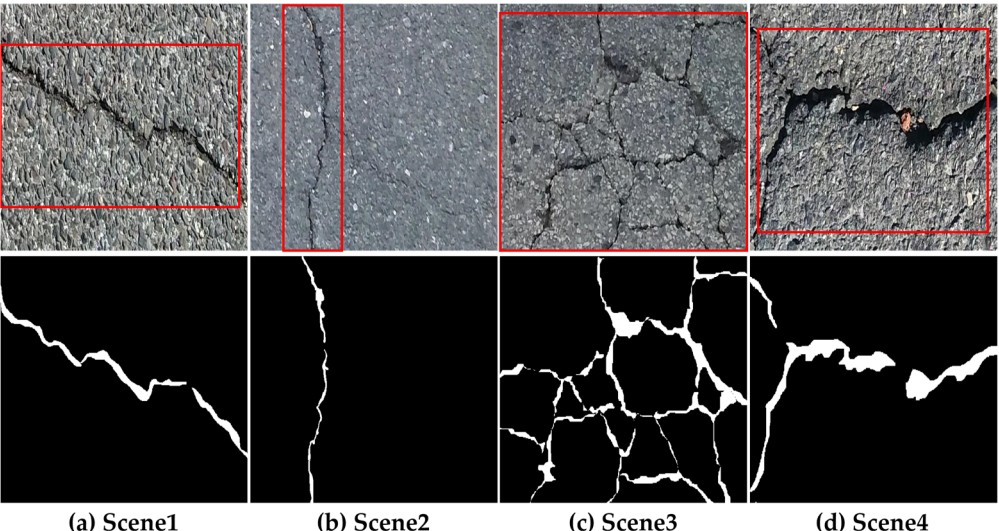

| (a) Scene1 | (b) Scene2 | (c) Scene3 | (d) Scene4 |

**Figure 5.** CRACK500 data sample.

### 3.2. Baselines and Implementation

This paper compares CAFANet with seven mainstream segmentation methods and find it to be effective, including the Deeply Supervised Image Fusion Network (DSIFNet) [48], DeepLabV3+ [49], Faster Fully Convolutional Network (FastFCN) [50], U-shaped Network (UNet) [23], Fully Convolutional Networks (FCN32s [22] and FCN8s [22]), Pyramid Scene Parsing Network (PSPNet) [51], Learning UNet (LUNet) [52], and Mixed Transformer UNet (MTUNet) [53]. Among them, DSIFNet and PSPNet are small-scale models, typically characterized by smaller network sizes and faster inference speeds. On the other hand, the other models are large-scale models with intricate network structures, allowing them to learn a greater number of potential semantic features. Taking the above-mentioned methods as reference points, we conduct a comprehensive evaluation of their performance. Additionally, to validate the approach, we conduct ablation experiments.

### 3.3. Visual Performance

3.3.1. Detection Results

Figure 6 presents some segmentation result examples of CAFANet, showcasing its excellent visual perception performance. The first, second, and third rows are the different segmentation results of four road damage images. The source image is (a), the label image is (b), and the segmentation result is (c). A red box marks the crack region in the original image. Furthermore, all the images show correctly segmented road damage regions at different scales, which indicates that the combination of the CLI module and FAB facilitates the integration of local and global information and enhances CAFANet's understanding of the spatial context and overall damage patterns. The CLI module allows CAFANet to capture long-range dependencies and context information, while the FAB ensures that CAFANet can properly integrate features from different branches and levels. This comprehensive feature fusion enables the model to make accurate predictions and generate more informative result images. In addition, CAFANet performs well in RCD tasks.

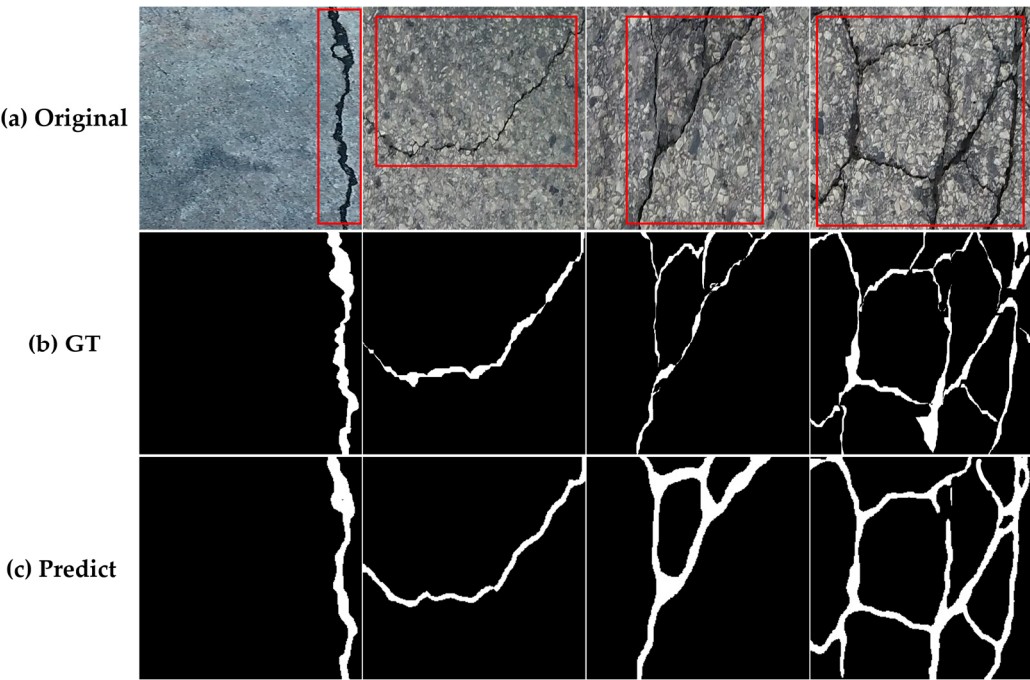

**Figure 6.** A few examples of the proposed method's segmentation results.

### 3.3.2. Attention Maps

The proposed CAFANet effectively identifies a diverse range of road damage images in the RCD task. As shown in Figure 7, to further demonstrate the superior performance of CAFANet, the CLI module's four stages are visualized and analyzed using attention images. Lower attention values are blue and higher attention values are red. A red box marks the crack region in the original image.

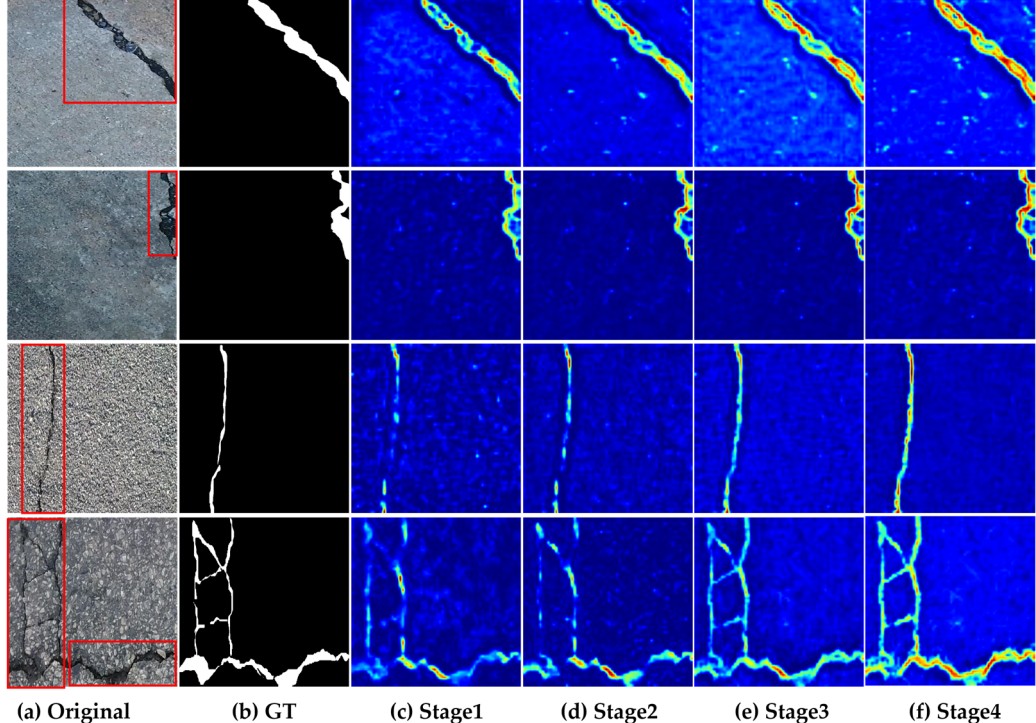

**Figure 7.** Visualization of attention maps in CRACK500 dataset.

Firstly, we analyze the images from a low level, which is in the first stage. By observing the corresponding attention maps, we can see that CAFANet primarily focuses on the edge and texture details in the images during this stage. At this point, most of the colors in the image appear as blue, indicating that the CLI module effectively captures subtle features in road damage images and utilizes them for subsequent processing. Next, we move on to the second stage, which is responsible for feature fusion and contextual understanding of the images. Analyzing the corresponding attention maps, we find that CAFANet directs its attention towards the contextual information surrounding the road damage regions. Moreover, at this stage, there is gradually less red in the damaged edge regions of the image. The CLI module's awareness enables CAFANet to better understand the location and boundaries of road damage, thereby improving the accuracy of detection. The third stage is the crucial detection stage, where CAFANet predicts the presence and accurate location of road damage. By observing the corresponding attention maps, we can see that as we move towards the center of the damaged region, a faint red color gradually appears. This indicates that CAFANet focuses its attention on the regions that potentially contain road damage during this stage. Lastly, we enter the fourth stage, which is the post-processing stage used to further optimize and refine the detection results. In this stage, the damaged regions in the image exhibit a deeper red color with a larger area. By analyzing the corresponding attention maps, we can observe that the proposed CAFANet guides attention towards crucial regions that contribute significantly to the road crack detection task. By doing so, it will be possible to detect damage with even greater precision and accuracy.

As we analyze the attention maps produced at each stage of the CLI module, we can clearly see that CAFANet performs better in detecting road damage images. It effectively captures fine-grained features and contextual information, accurately locates road damage, and enhances the results through post-processing. These characteristics enable CAFANet to achieve outstanding performance and robustness in road damage image detection. The research results demonstrate the effectiveness of CAFANet and give strong support to the idea that road damage can be detected and identified quickly and accurately.

### 3.3.3. Comparative with Baselines

In this section, we conduct a comparison between our approach and a baseline method. Figures 8–16 show the results of segmentation for road damage by various methods. We mark the transverse cracks in the original image with a red box, while the white outlines of the label and segmentation maps represent the damaged region of the road, and the black region forms the background. Among them, Figures 8–10 show the comparison results between CAFANet and the comparison method with segmental transverse cracks of the road, while Figures 11–13 show the comparison results between CAFANet and the comparison method with segmental longitudinal cracks of the road. Figures 14–16 show the comparative results between CAFANet and the comparison method in segmental road mesh cracks. Figures 8–16 show that using the CLI module and FAB produces a good segmentation result, while other methods perform slightly worse, indicating they approximate the effective characteristics of road damage segmentation. A multi-scale feature interactive learning approach is more effective at segmenting road damage than other large-scale models, indicating that CAFANet provides better segmentation than the mainstream models. We can see that the segmentation results of CAFANet are significantly more complete than those obtained by other methods. In addition, CAFANet makes it possible to segment the image more accurately than other methods, with most of the damaged regions being correctly segmented. This indicates that the CLI module and FAB are helpful in improving CAFANet's feature extraction and recognition. To summarize, the visual results depicted in Figures 8–16 provide clear evidence that CAFANet excels in segmenting road damage images, outperforming the baseline methods to some extent.

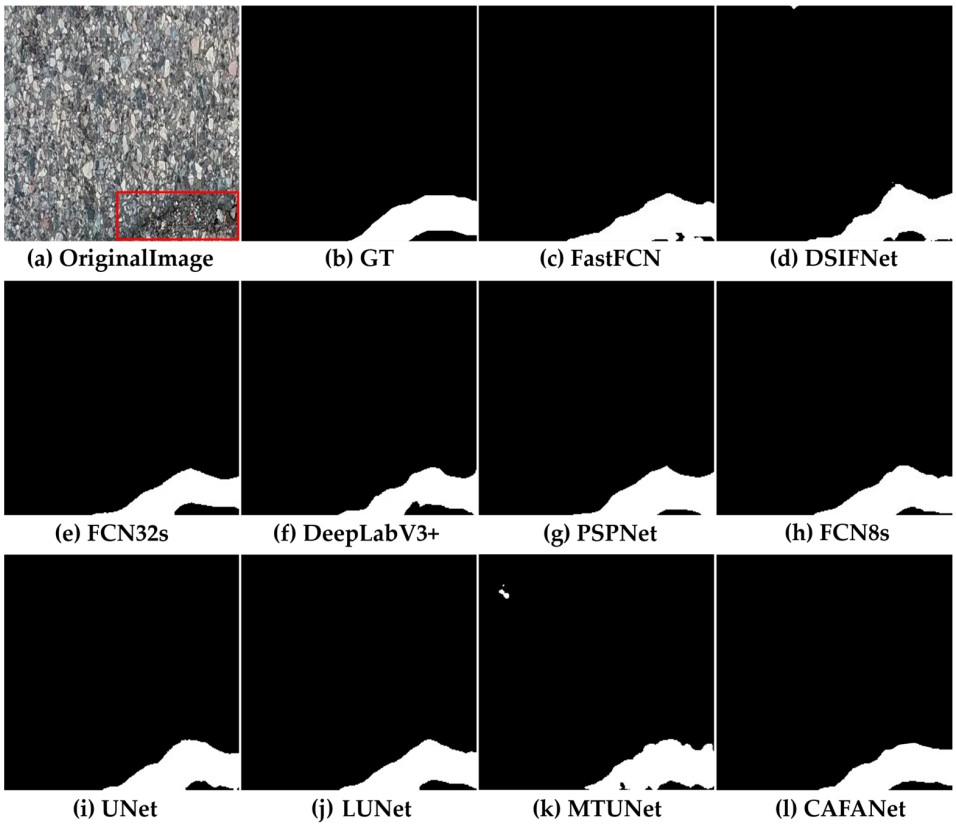

**Figure 8.** Segmentation result examples obtained by various methods for transverse cracks.

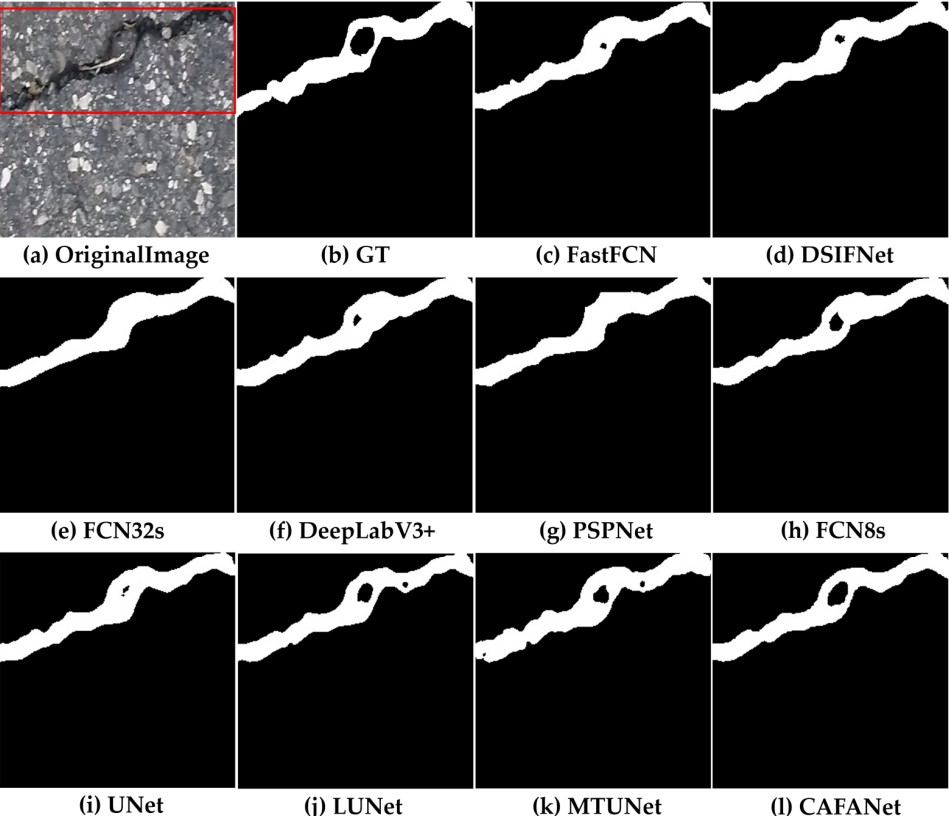

**Figure 9.** Segmentation result examples obtained by various methods for transverse cracks.

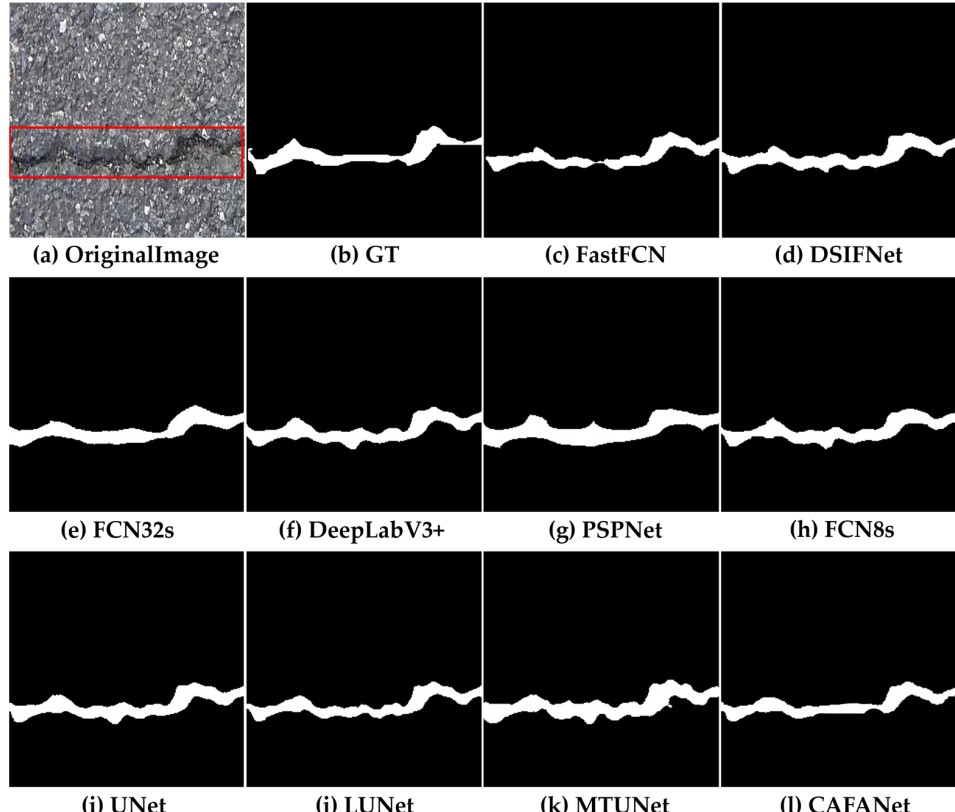

**Figure 10.** Segmentation result examples obtained by various methods for transverse cracks.

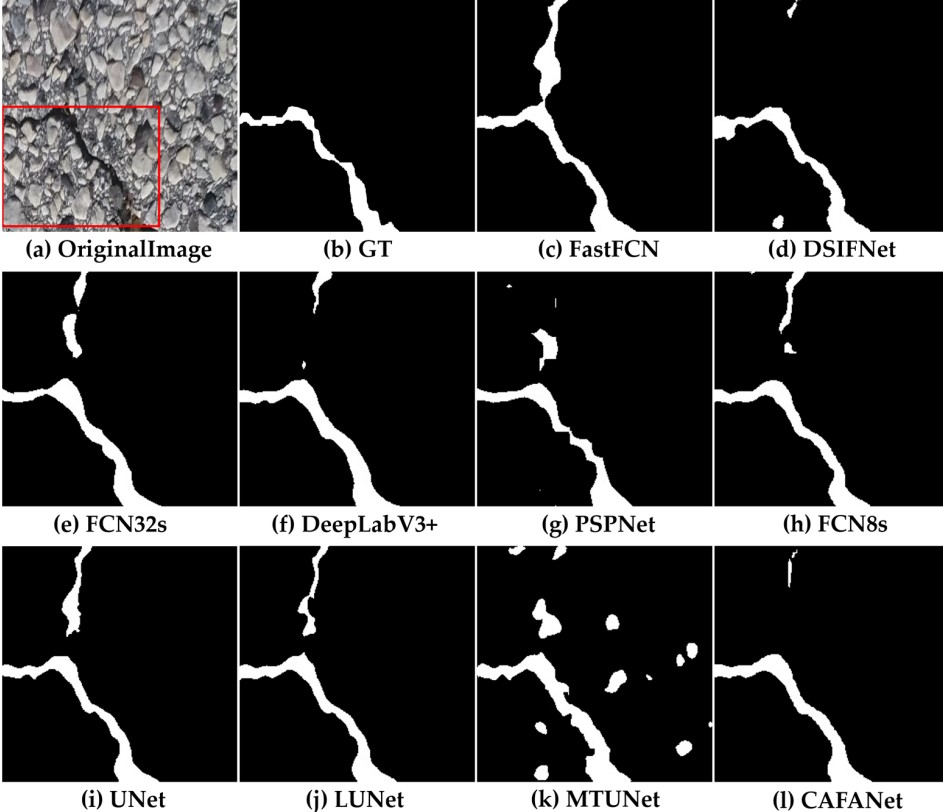

**Figure 11.** Segmentation result examples obtained by various methods for longitudinal cracks.

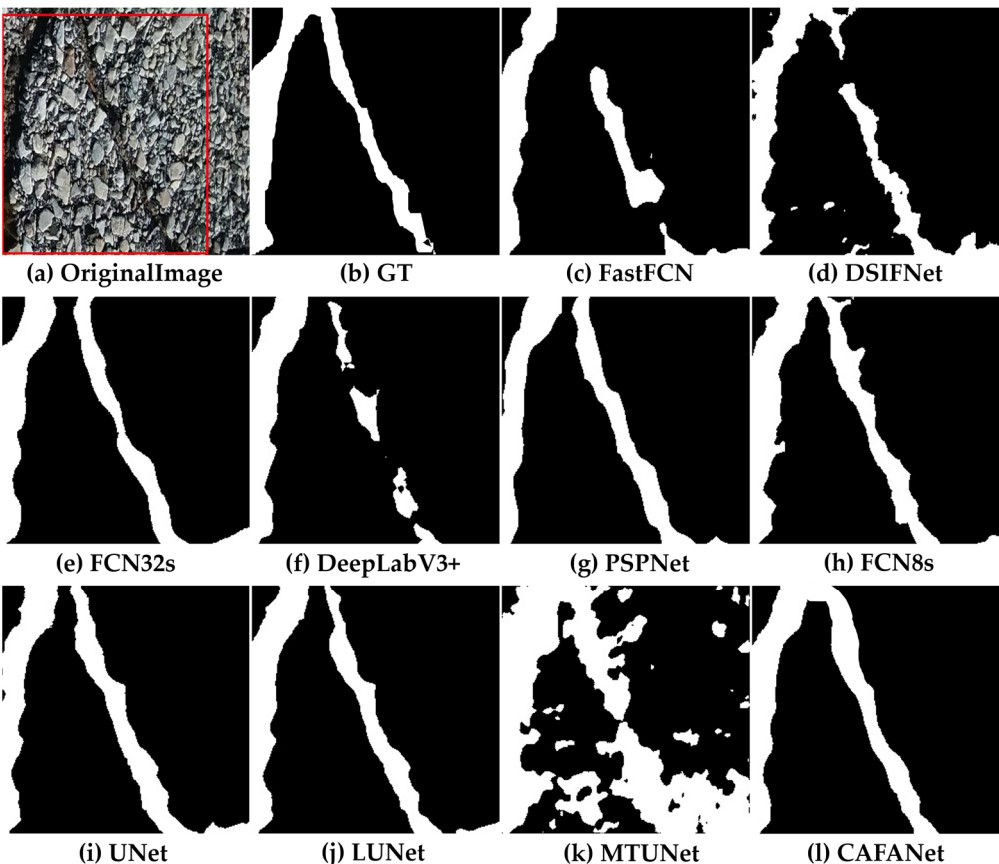

**Figure 12.** Segmentation result examples obtained by various methods for longitudinal cracks.

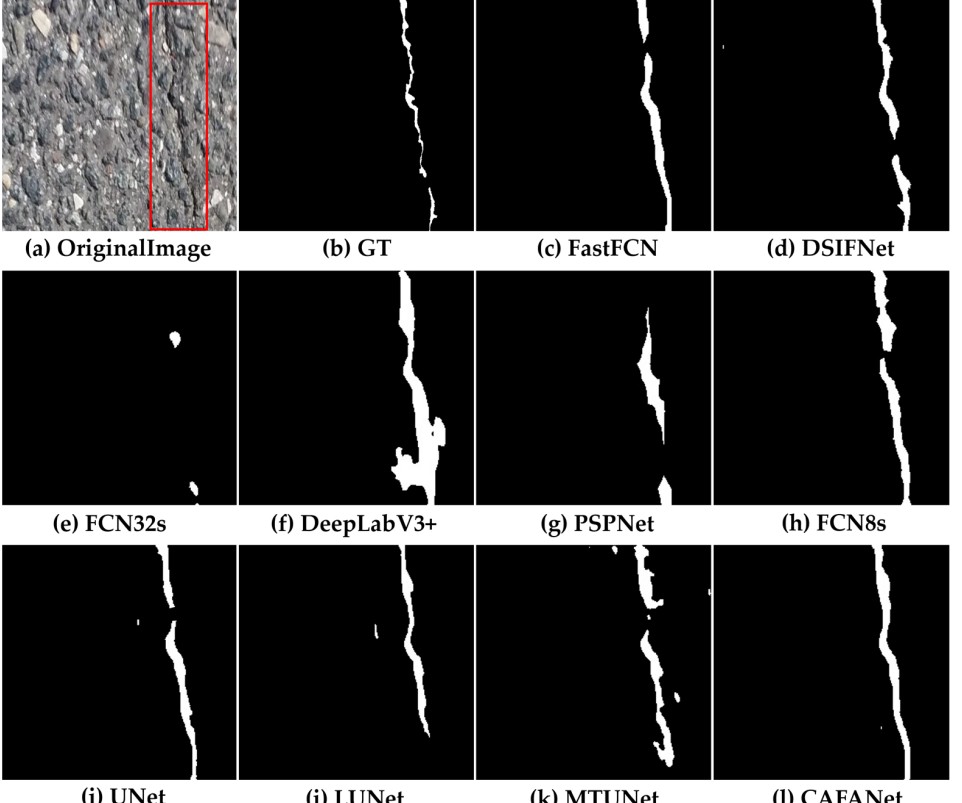

**Figure 13.** Segmentation result examples obtained by various methods for longitudinal cracks.

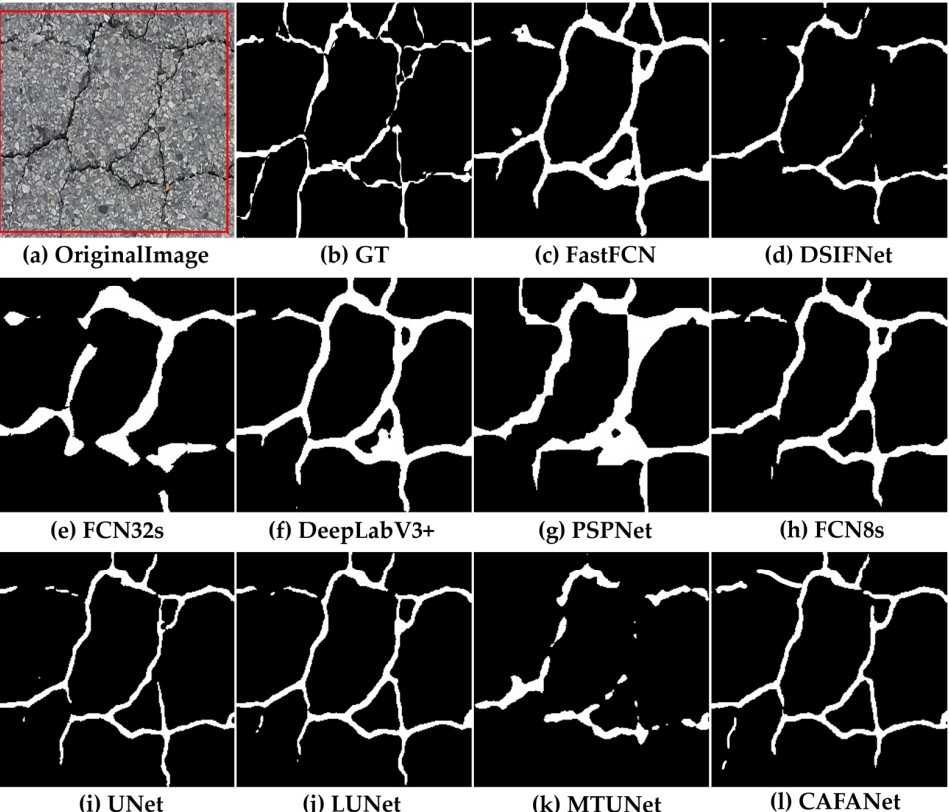

**Figure 14.** Segmentation result examples obtained by various methods for mesh cracks.

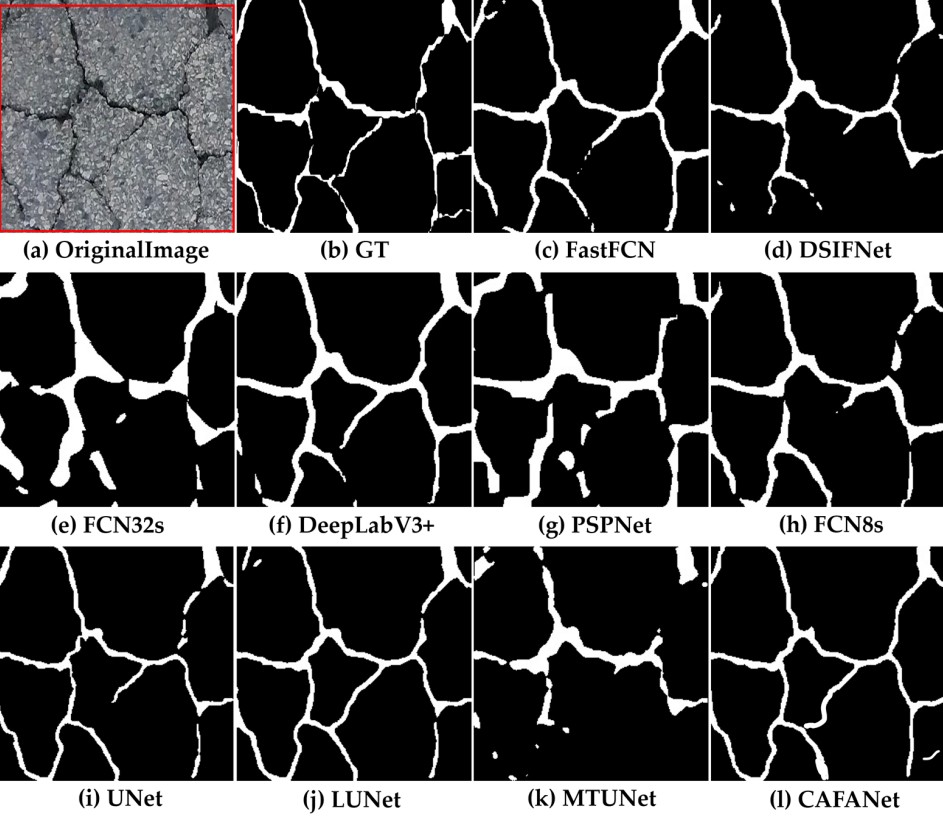

**Figure 15.** Segmentation result examples obtained by various methods for mesh cracks.

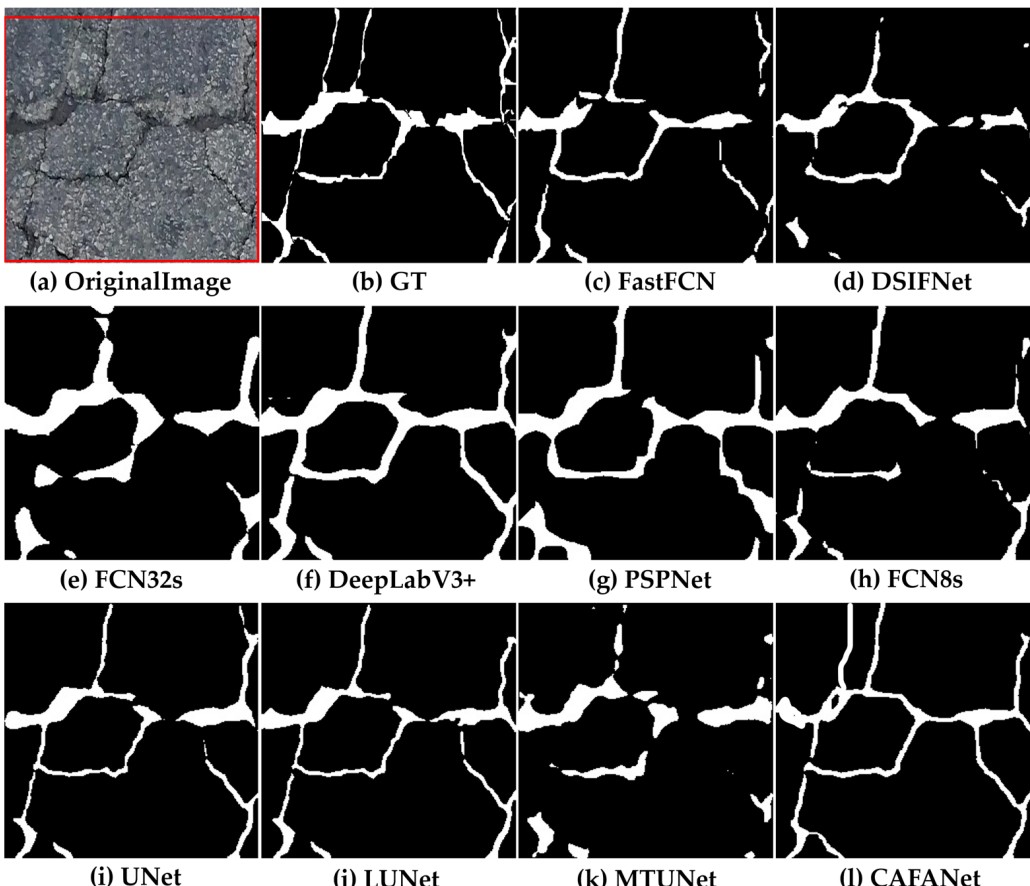

**Figure 16.** Segmentation result examples obtained by various methods for mesh cracks.

### 3.4. Quantitative Analysis

This paper presents a quantitative comparison of CAFANet with the baseline methods in Table 1. As we can see, CAFANet achieves 68.46% in *Precision*, 73.22% in *F1 score*, 96.63% in *IOU_0*, 57.75% in *IOU_1*, 77.19% in *mIOU*, 96.78% in *OA*, and 71.52% in *Kappa*. Among the baseline methods, DSIFNet obtains the highest *Precision* of 67.42%, while FastFCN achieves the highest values in six metrics: *F1 score* (72.52%), *IOU_0* (96.51%), *IOU_1* (56.88%), *mIOU* (76.7%), *OA* (96.66%), and *Kappa* (70.75%). It can be observed that among the seven indices, CAFANet's *Precision* is 1.04% higher than that of DSIFNet, and the other six indices of CAFANet are 0.7%, 0.12%, 0.87%, 0.49%, 0.12%, and 0.77% higher than those of FastFCN.

**Table 1.** The quantitative results of proposed method on CRACK500.

| Method | Precision | Recall | F1 Score | IOU_0 | IOU_1 | mIOU | OA | Kappa |
|---|---|---|---|---|---|---|---|---|
| UNet | 66.41 | 79.44 | 72.34 | 96.45 | 56.67 | 76.56 | 96.6 | 70.55 |
| LUNet | 67.41 | 78.93 | 72.71 | 96.54 | 57.13 | 76.83 | 96.69 | 70.96 |
| MTUNet | 47.17 | 80.3 | 59.43 | 93.58 | 42.28 | 67.93 | 93.87 | 56.35 |
| FastFCN | 67.19 | 78.76 | 72.52 | 96.51 | 56.88 | 76.7 | 96.66 | 70.75 |
| DeepLabV3+ | 61.42 | 83.37 | 70.73 | 95.95 | 54.72 | 75.34 | 96.14 | 68.72 |
| DSIFNet | 67.42 | 73.55 | 70.35 | 96.39 | 54.27 | 75.33 | 96.53 | 68.52 |
| FCN32s | 61.81 | 78.81 | 69.27 | 95.91 | 53.01 | 74.46 | 96.09 | 67.23 |
| FCN8s | 66.55 | 78.93 | 72.21 | 96.45 | 56.51 | 76.48 | 96.6 | 70.42 |
| PSPNet | 61.26 | 79.74 | 69.29 | 95.86 | 53 | 74.43 | 96.05 | 67.21 |
| CAFANet | 68.46 | 78.69 | 73.22 | 96.63 | 57.75 | 77.19 | 96.78 | 71.52 |

Note that the units of all indicators are percentages. The best results in red, the next best in blue.

Compared with the baselines, CAFANet achieves leadership in almost all metrics for RCD tasks except *Recall*, which is slightly lower. More specifically, the maximum *Precision*

means that a higher percentage of the predicted results of CAFANet are actually road damage, reducing the possibility of false positives, the best *F1 score* expresses that CAFANet is better able to capture true road damage, reducing the possibility of missed detection, the highest *IOU_0* shows that CAFANet can accurately distinguish between road damage and background, the best *IOU_1* value indicates that CAFANet effectively captures the location and shape of road damage, the highest *mIOU* illustrates that CAFANet can accurately predict regions of damage category, the highest OA declares CAFANet to be more accurate in predicting road damage overall, and the highest *Kappa* implicates that the predicted results of CAFANet have a high consistency with the actual labeling, which reduces the influence of random errors. Although the *Recall* of the proposed method scheme lags behind other methods, it makes sense because CAFANet selects more significant regions of the source image and ignores those that are less critical. In a more specific sense, CAFANet leverages comprehensive global interactions within road damage images. It prioritizes the prominent regions, leading to information loss in less significant areas.

As an additional measure highlighting the exceptional performance of the proposed method, we perform segmentation detection on 141 images using CAFANet and baseline methods. It is shown in line graphs in Figure 17 for four metrics: *F1 score*, *mIOU*, *Precision*, and *IOU_1*. A curve with points (x, y) indicates that 100% * x % of images have metrics that do not exceed y. By comparative analysis of these metrics, we can evaluate the performance of different methods for road crack detection and determine whether CAFANet is optimal.

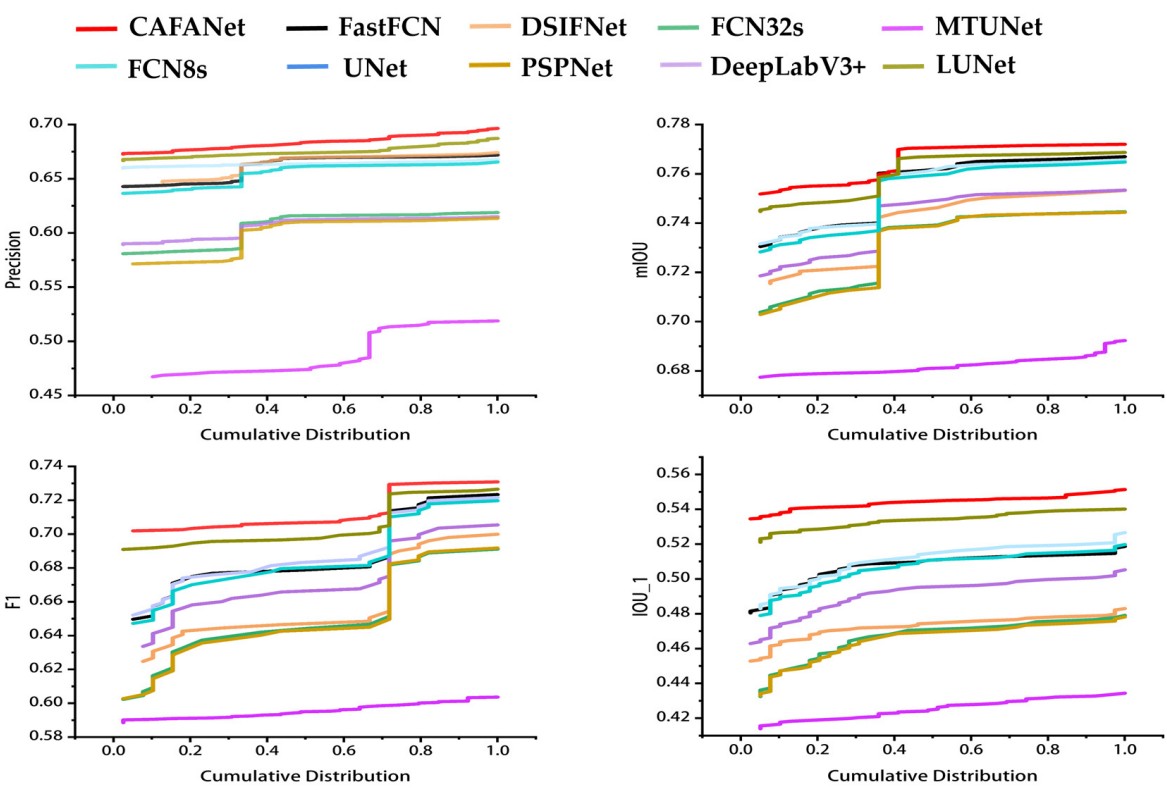

**Figure 17.** The quantitative comparison of *Precision*, *IOU_1*, *F1 score*, and *mIOU* metrics.

First, observe the line graph of *F1 score*. It utilizes *Precision* and *Recall* to measure classification accuracy and omission rate. By comparing the *F1 score* curves of different methods, we can see that CAFANet achieves the highest *F1 score* on most images. This demonstrates that CAFANet strikes the optimal balance between classification accuracy and omission rate in road crack detection. Next, let us focus on the line graph of *mIOU*, which measures the model's prediction accuracy in multi-class problems. By comparing the *mIOU* curves of different methods, we find that CAFANet has the highest *mIOU*

value on most images. This means that CAFANet can accurately predict the location of road damage regions and has a high degree of overlap with the actual annotations. In addition, the line graphs of *Precision* and *IOU_1* are also significant references for evaluating model performance. *Precision* reflects the proportion of samples that are predicted by the model to be positive among the true positive samples, while *IOU_1* represents the overlap between the predicted road damage regions and the actual annotations. By comparing the *Precision* and *IOU_1* curves of different methods, the results of these two metrics show that CAFANet is also the most effective. In conclusion, through the comparative analysis of the line graphs of *F1 score*, *mIOU*, *Precision*, and *IOU_1*, CAFANet delivers better results in road crack detection.

*3.5. Ablation Anaysis*

As shown in Table 2, this paper presents the results of the FAB and CLI module performance evaluation on the CRACK500 dataset.

**Table 2.** Results of ablation experiments using CRACK500 dataset.

| Method | Precision | Recall | F1 Score | IOU_0 | IOU_1 | mIOU | OA | Kappa |
|---|---|---|---|---|---|---|---|---|
| Baseline | 68.87 | 76.73 | 72.58 | 96.61 | 56.97 | 76.79 | 96.76 | 70.87 |
| Baseline + CLI | 66.33 | 80.52 | 72.74 | 96.47 | 57.16 | 76.81 | 96.63 | 70.96 |
| Baseline + FAB | 68.29 | 78.76 | 73.15 | 96.62 | 57.67 | 77.15 | 96.77 | 71.44 |
| Baseline + CLI + FAB | 68.46 | 78.69 | 73.22 | 96.63 | 57.75 | 77.19 | 96.78 | 71.52 |

Upon adding the CLI module separately, the *F1 score* increases to 72.74% compared to the baseline's 72.58%. Similarly, only by adding the FAB, the *F1 score* grows to 73.15% compared to the baseline. When we simultaneously incorporate the CLI module and FAB, the model achieves an *F1 score* of 73.22%, which is a 0.64% improvement over the baseline. If we only have the CLI module, *Kappa* shows a modest improvement of 0.09%, while the FAB leads to a 0.57% improvement in *Kappa*. This indicates that the FAB enhances the prediction results, making them more consistent with the actual situation. Finally, by integrating all modules, the *F1 score*, *IOU_1*, *mIOU*, *OA*, and *Kappa* for RCD reach their highest values of 73.22%, 57.75%, 77.19%, 96.78%, and 83.37%, respectively. This surpasses the baseline's performance.

**4. Case Study**

Road damage can occur on different types of roads, with varying degrees of damage, and under various weather and lighting conditions. Road crack detection is a crucial task related to road safety, and the accuracy and reliability of the model are crucial for practical applications. To evaluate and validate the generalization ability of CAFANet in different environments and conditions, we conduct generalization experiments on the public dataset CPRID. According to the experimental results, CAFANet performs well and is applicable to new data.

*4.1. Datasets*

The Cracks and Potholes in Road Images Dataset (CPRID) [54] serves as the primary dataset for this research, offering a comprehensive and widely utilized dataset specifically designed for road crack detection and analysis. It consists of a diverse collection of high-resolution road images annotated with cracks and potholes. The CPRID contains approximately 2235 images. We partition it into training, validation, and test sets in a ratio of 5:1:4, similar to the division in CRACK500. In total, we obtain a training set with 1118 images, a verification set with 223 images, and a test set with 603 images. In the road damage images, the pixel values corresponding to the length and width of potholes are also different. According to the label image of the source image (the white area is the pothole, the background area is black), the corresponding pothole pixel value can be calculated. Finally, through a comparative analysis of the sizes of all pixel values, potholes

with pixel values below 10,000 pixels are defined as small potholes, those between 10,000 and 30,000 pixels are defined as medium potholes, and those above 30,000 pixels are defined as large potholes. Figure 18 shows some examples of potholes.

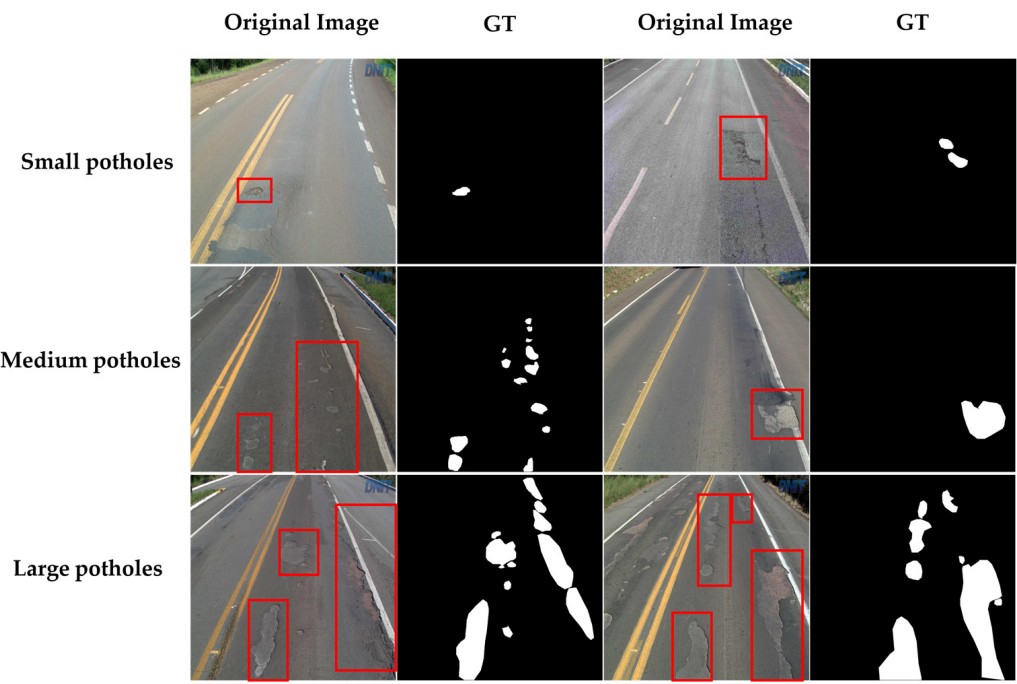

**Figure 18.** CPRID data sample.

*4.2. Detection Results*

We extensively evaluate CAFANet on the CPRID, demonstrating its robust generalization performance across various pothole sizes. Figure 19 presents a subset of source images along with their corresponding ground truth labels and detection results. The first, second, and third rows are the different segmentation results of four road damage images. In the original image, a red box marks the pothole region. A visual analysis reveals that the network detects small, medium, and large potholes well.

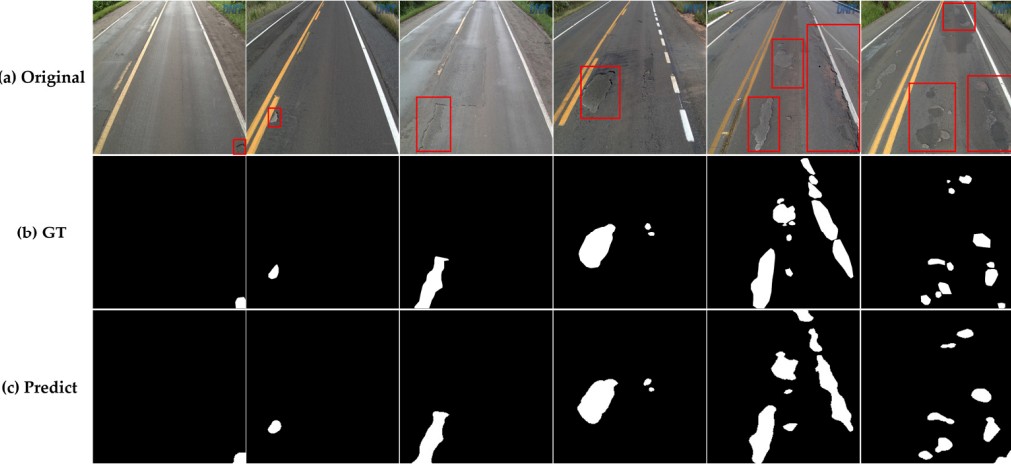

**Figure 19.** CPRID segmentation result examples.

For small potholes, CAFANet effectively captures their subtle features and correctly localizes them within the image. This precise detection is crucial for identifying potential road hazards and timely repairs. CAFANet demonstrates reliable performance in medium pothole detection and segmentation. It accurately outlines the boundaries of potholes and

provides detailed pixel-level segmentation, enabling a comprehensive understanding of the pothole's extent and shape. Even for large potholes, CAFANet detects and segments them. The proposed method successfully captures significant features and accurately distinguishes large potholes from the surrounding road surface. The combination of the CLI module and FAB allows CAFANet to leverage contextual information and capture multi-scale features. This capability enables the network to handle variations in pothole sizes effectively and consistently, leading to strong performance across the entire range of pothole sizes.

Overall, experimental results on the CPRID demonstrate the impressive generalization capability of the network that combines cross-attention and feature alignment. It accurately detects and segments small, medium, and large potholes, showcasing the network's reliable performance across different pothole sizes.

## 5. Conclusions

The detection of road damage is an important part of road maintenance and safety. In recent years, researchers have explored numerous methods to enhance the accuracy and efficiency of road crack detection systems. This paper proposes a CAFANet architecture for detecting road damage. Firstly, by incorporating the CLI module into CAFANet, we aim to leverage its advantages for precise and reliable detection of damaged areas. The CLI module effectively exchanges information between different levels or branches of the neural network, promoting enhanced feature learning and representation, thus improving CAFANet's multi-scale feature expression capability. Secondly, we employ the FAB to address challenges arising from variations in road conditions, lighting, and background textures. In road crack detection, aligning features at different scales and levels is crucial for capturing comprehensive information and accurately representing damaged regions. Through feature alignment, CAFANet effectively distinguishes critical information from irrelevant details and focuses more on regions of interest, thereby enhancing its detection capability for small targets. We conduct experiments on the CRACK500 dataset, and the results demonstrate CAFANet's effectiveness in terms of detection accuracy for road damage.

This paper emphasizes the importance of integrating the CLI module and FAB technique in road crack detection. The CLI module enables CAFANet to capture long-range dependencies and effectively utilize contextual information. The FAB operation further enhances the model's alignment and integration capability for features at different scales, improving the comprehensive understanding of damaged regions. By combining these techniques, the model can improve contextual accuracy, align features at different scales, and accurately detect and localize road damage. In the real world, road cracks are usually a minority category. The imbalance of sample categories may cause the model to pay too much attention to most categories, thus affecting the detection performance for road cracks. Future research can further explore the optimization and extension of the CLI module and FAB technique for road crack detection. This will be carried out by applying them to practical applications to assist transportation departments in effectively maintaining roads.

**Author Contributions:** Conceptualization, Chuan Xu, Qi Zhang, Liye Mei, and Wei Yang; methodology, Qi Zhang; software, Xiufeng Chang; validation, Liye Mei, Zhaoyi Ye, and Qi Zhang; formal analysis, Wei Yang; investigation, Liye Mei; data curation, Junjian Wang and Lang Ye; writing—original draft preparation, Qi Zhang; writing—review and editing, Chuan Xu; visualization, Liye Mei; supervision, Chuan Xu, Wei Yang, and Xiufeng Chang; project administration, Chuan Xu; funding acquisition, Chuan Xu and Wei Yang. All authors have read and agreed to the published version of the manuscript.

**Funding:** Scientific Research Foundation for Doctoral Program of Hubei University of Technology (BSQD2020056); Natural Science Foundation of Hubei Province (2022CFB501): University Student innovation and Entrepreneurship Training Program Project (202210500028).

**Data Availability Statement:** Not applicable.

**Conflicts of Interest:** The authors declare no conflict of interest.

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
