# Peer review of "Cross-Attention-Guided Feature Alignment Network for Road Crack Detection"

_ijgi, doi:10.3390/ijgi12090382_

Round 1
Reviewer 1 Report (Previous Reviewer 1)
Comments and Suggestions for Authors
Content from the review literature has been done to an extent (90%) and can help in future research directions, The novelty aspects of the contribution over other approaches to the same problem are explained clearly.
Author Response
Thank you very much for your modification suggestions. Please see the attachment for the specific reply.

Reviewer 2 Report (Previous Reviewer 2)
Comments and Suggestions for Authors
Please refer to the attachment.

Moderate editing of English language required
Author Response
Thank you very much for your modification suggestions. Please see the attachment for the specific reply.

Reviewer 3 Report (New Reviewer)
Comments and Suggestions for Authors
The author proposes a road damage detection method and validates its effectiveness through experiments on a publicly available dataset. However, there are several issues that the author should carefully consider:
1. Could the title of the paper be revised to "Road Crack Detection" instead of "Road Damage Detection"?
2. The comparative methods in this paper include classic semantic segmentation networks like FCN, U-Net, and Deeplab. These methods seem outdated in comparison to the strategies employed in this paper, such as attention mechanisms. It is recommended to introduce improved methods like attention-based modifications to U-Net for a more relevant comparison.
3. Based on the results of ablation experiments, the improvement brought about by the proposed strategies in this paper appears marginal, especially given the small dataset. There might be concerns about overfitting the model. It's advisable to conduct comparative experiments on a larger dataset to assess the effectiveness more robustly.
4. The red bounding boxes used to indicate road damage in the visualized images might be unnecessary.
Author Response
Thank you very much for your modification suggestions. Please see the attachment for the specific reply.

Round 2
Reviewer 2 Report (Previous Reviewer 2)
Comments and Suggestions for Authors
The authors have revised all the comments. I have no further comments and suggestions.
Comments on the Quality of English LanguageMinor English language editing is required due to extra spaces and spelling mistakes.
Reviewer 3 Report (New Reviewer)
Comments and Suggestions for Authors
There is no comment.
This manuscript is a resubmission of an earlier submission. The following is a list of the peer review reports and author responses from that submission.
Round 1
Reviewer 1 Report
Comments and Suggestions for Authors
This paper introduces A Cross Attention Guided Feature Alignment Network for Road Damage Detection.
Overall, the manuscript proposes a cross-attention guided feature alignment network (CAFANet) for extracting and integrating multi- scale features of road damages.
The article needs to undergo the following revisions in different sections.
11. Each acronym should be explained the first time it appears in the text, even if it appeared in the abstract. Check all abbreviations in text: each word should start with capital to explain an abbreviation, All keywords (abbreviations) must be mentioned in the abstract.
2. The paper should focus on a short paragraph to introduce what the rest of the paper contents will follow at the end of the Introduction section is missing. This paragraph is important; as it can enable the readers to understand what the following content will be and arouse their interest to continue reading the paper.
32. All keywords (abbreviations) must be mentioned in the abstract.
43. Need clarity in introduction and related work sections- the ambiguity has been found.
54. Redraft all the titles and contents appropriately- ref section 2 – materials and methods to be more clear- fuzzy in nature.
65. In Section 2, ref fig 1 title need to be precise not as a paragraph, this holds good in section 3 also for all the figures as illustrated.
76.The experimental results in a tabular form shall be easier to understand the result outcomes than a paragraph to avoid confusion.
87. Further, as per the article contents there still exists revision in English grammar and word choice as used- use of we at multiple sections to be avoided.
Comments on the Quality of English Languageas per the article contents there still exists revision in English grammar and word choice as used- use of we at multiple sections to be avoided.
Reviewer 2 Report
Comments and Suggestions for Authors
Please find the attachment.

Reviewer 3 Report
Comments and Suggestions for Authors
1. Abbreviations should be defined before using, e.g., RDD, VOC and etc.
2. Which performance is high for the methods presented in [8] and [9]? Moreover, which performance is limited as mentioned in line 56?
3. Ref. [11] also uses the random forest to realize RDD. Then why it belongs to advanced machine learning techniques, but [9] not as mentioned in lines 54-55 and 60?
4. In line 61, what does “Overall, the above … ” mean?
5. It is better to put the content in lines 62-66 at the beginning, since it seems that the methods presented in [7]-[11] can detect automatically.
6. The content in lines 76-100 should be re-written to organize the state of the art, not just list the references.
7. The writing should be improved greatly, e.g., inconsistent variable style, many grammar mistakes and etc.
Comments on the Quality of English LanguageThe writing should be improved greatly.
